# Betaine Alleviates LPS-Induced Chicken Skeletal Muscle Inflammation with the Epigenetic Modulation of the TLR4 Gene

**DOI:** 10.3390/ani12151899

**Published:** 2022-07-26

**Authors:** Feng Guo, Mengna Jing, Aoyu Zhang, Yan Yu, Pei Gao, Qiuxia Wang, Li Wang, Zhiyong Xu, Jinyou Ma, Yanhong Zhang

**Affiliations:** College of Animal Science and Veterinary Medicine, Henan Institute of Science and Technology, Xinxiang 453003, China; Guofeng@hist.edu.cn (F.G.); 18738375527@163.com (M.J.); zhangaoyu1111@163.com (A.Z.); yymtt03@163.com (Y.Y.); peipei594@163.com (P.G.); wqxmjz@126.com (Q.W.); wangli19841027@163.com (L.W.); xuzhiyong_0_817@126.com (Z.X.); marsjy@163.com (J.M.)

**Keywords:** betaine, skeletal muscle, methylation, TLR4, IL6

## Abstract

**Simple Summary:**

The poultry meat we eat is the skeletal muscle which comprises approximately three-quarters of the body weight of a chicken. In the modern poultry industry, the intensively raised broilers face the risk of exposure to environmental factors which can cause acute or chronic systemic inflammation. Inflammation, in return, contributes to the pathology of skeletal muscle diseases which are characterized by the loss of skeletal muscle mass. By adding betaine, a natural component, into the water of the newly hatched broilers for two weeks, we found that inflammation-related gene expression in the leg muscle was remarkably reduced. Specifically, we found that betaine inhibited the LPS-induced abnormal expression of IL-6 and TLR4. Further study indicated that the methylation modulation of the gene may be involved in betaine’s action. We suggest that betaine could be considered a safe and cheap preventive reagent candidate for chicken skeletal muscle inflammatory diseases.

**Abstract:**

Betaine was found to alleviate inflammation in different studies. Here, newly hatched broilers were randomly divided into control and betaine consumptive groups, who had access to normal drinking water and water with betaine at a dose of 1000 mg/L, respectively. At the age of two weeks, the boilers were intraperitoneally treated with LPS. The protective effects of betaine against LPS-induced skeletal muscle inflammation were studied. Betaine attenuated the LPS-induced overexpression of IL-6 significantly in the leg muscle. Furthermore, LPS lowered the expression of TLR4 and TLR2 but increased the expression of MyD88. Betaine eliminated the effect of LPS on the expression of TLR4 but not TLR2 and MyD88. LPS also increased the expression of Tet methylcytosine dioxygenase 2 (Tet2), and this effect was also eliminated by betaine consumption. MeDIP-qPCR analysis showed that the methylation level in the promoter region of IL-6 was decreased by LPS treatment, whilst betaine cannot prevent this effect. On the contrary, LPS significantly increase the methylation level in the promoter region of TLR4, which was decreased by the consumption of betaine. Our findings suggest that betaine can alleviate LPS-induced muscle inflammation in chicken, and the regulation of aberrant DNA methylation might be a possible mechanism.

## 1. Introduction

Inflammation is the result of stress, regardless of the source or nature of the stress (biological, environmental, nutritional, physical, chemical, or psychological). Poultry production has experienced a major shift with the prohibition of the use of antimicrobials and antibiotics as growth promoters, which may lead to an increase in the inflammatory level [1]. Inflammation, in return, contributes to several pathologies such as skeletal muscle diseases which are characterized by the loss of skeletal muscle mass and the reduction in poultry performance [2]. Betaine was originally known to be an important osmoprotectant and also an important methyl donor in the methionine cycle of cells. Accumulating evidence also showed that betaine supplemented in the diet plays a role in the anti-inflammation process [3]. In mammals, dietary betaine supplementation has been reported to be protective in different types of inflammatory circumstances such as high-fat diet-induced adipose inflammation [4], ischemia/reperfusion-induced brain inflammation [5], high-fructose-induced renal inflammation [6], and the inflammation state in the liver in non-alcoholic diseases [7].

In the animal feed industry, betaine is also a dietary component with well-reported hepatoprotective effects. Studies by Zhao et al. showed that the maternal betaine supplementation or egg injection of betaine were efficient to protect the offspring chickens from high hepatic cholesterol in different circumstance [8,9,10]. In addition, beneficial effects of dietary betaine on egg-laying performance, growing performance, meat quality, and anti-oxidation have also been reported [11,12,13,14]. However, as a natural compound, the role of betaine in anti-inflammation in domestic animal husbandry has not been well revealed. Limited studies showed that dietary betaine was able to alleviate the hot stress-induced mucosal inflammation in broilers by lowering the expression of interleukin (IL) 1β and the 70 kDa heat shock protein (HSP70) mRNA and increasing the expression of IL-10 [15]. Betaine supplemented in drinking water was able to alleviate the IBDV-induced bursal inflammation by inhibiting the overexpression of IL-6 and type I interferon (IFN) [16]. An in vitro study showed that LPS-induced IL-6 overexpression in porcine intestinal epithelial cells was alleviated by supplying betaine into the media [17]. Inflammation is an important contributor to the pathology of skeletal muscle diseases which are characterized by the loss of skeletal muscle mass [18]. LPS is a component of gram-negative bacteria, which always threaten intensively raised animals, causing acute or chronic systemic inflammation. Whether betaine has an anti-inflammatory function in LPS-induced broiler muscle inflammation has not been reported. 

Betaine is a methyl donor. Dietary betaine can provide the methyl group via the methionine cycle, which produces S-adenosylmethionine, the principal methylating agent [19]. The methyl groups provided by the donors can be used in the process of DAN methylation, which is dynamically regulated through the action of the DNA methyltransferase (DNMT) and demethylating proteins, including the ten-eleven translocation (TET) family of dioxygenases. As a well-known epigenetic modification, DNA methylation is involved in various biological processes by negatively regulating gene expression [20]. Studies carried out on both mammals and birds showed that betaine might regulate the expression of a wide range of genes involved in lipid and glucose homeostasis through “nutrient responsive” epigenetic mechanisms [8,21,22,23].

Previous studies showed that betaine can inhibit LPS-induced nuclear factor-κB (NF-κB) activation [24] and decrease high-fat-diet-induced hepatic toll like receptor (TLR) 4 mRNA expression [25], which contributes to its anti-inflammation function. However, the epigenetic mechanism involved in the anti-inflammatory activity of betaine is rarely reported. We previously showed that betaine supplementation through drinking water alleviated IBDV-induced IL-6 overexpression through increasing the methylation level in the promoter region [16]. Hence, in the present study, we aimed to evaluate the anti-inflammatory effects of betaine on LPS-induced muscle inflammation and the possible epigenetic mechanism.

## 2. Materials and Methods

### 2.1. Animals and Experimental Design

The experimental procedures and animal management in this study were approved by the Institutional Animal Care and Use Committee at the Henan Institute of Science and Technology. Forty commercial ross 308 eggs were obtained from Henan Doyoo Industrial Co., Ltd. (Hebi, China) and incubated in an electric forced-draft incubator at 37.5 ± 0.5 °C and 65% relative humidity. At the day of hatching, 30 chicks with similar body weights (BW) were randomly divided into three groups—the control group, LPS group and Betaine + LPS group—with an equal number in each group. The chickens were kept in one room at a standard temperature, moisture, light, and ventilation. Water and a corn-soybean meal diet were provided ad libitum. Betaine at a dose of 1000 mg/L was added to the drinking water of the Betaine+LPS group. After two weeks of feeding, the chickens in the LPS and Betaine+LPS groups were intraperitoneally injected with LPS from *Escherichia coli* 055:B5 (Sigma-Aldrich, Darmstadt, Germany) at a dose of 1 mg/kg BW. The chickens in the control group were intraperitoneally injected with the same volume of PBS. Two hours after the treatment, six chickens in each group were randomly chosen and sacrificed, and the leg muscles were collected, snap-frozen in liquid nitrogen, and stored at −70 °C.

### 2.2. Quantitation of mRNA Expression via Real-Time PCR

Muscle samples were ground in liquid nitrogen, and about 30 mg of each sample was used to extract the total RNA with a TRIzol reagent (Invitrogen, Carlsbad, CA, USA) according to the manufacturer’s instructions. The total RNA extracts were then treated with DNase I (Takara, Shiga, Japan) to eliminate the possible contamination of genomic DNA. The quantification and integrality of the RNA were checked with a NanoDrop ND-1000 instrument (Thermo Fisher Scientific, Wilmington, DE, USA) and formaldehyde-containing agarose electrophoresis, respectively. Two micrograms of the total RNA were subjected to reverse transcription to generate cDNA by using the GoScript Reverse Transcription System (Promega, Madison, WI, USA). A total of 2 μL of diluted cDNA (1:15) was subjected to real-time PCR using TB Green^®^ Premix Ex Ta II (TaRaKa, Shiga, Japan). All the primers were designed by using Primer Premier 5 software (Premier Biosoft, Palo Alto, CA, USA), and the sequence of the primers was listed in Table 1. Chicken β-actin was selected as a reference gene. 

### 2.3. Methylated DNA Immunoprecipitation Analysis

Methylated DNA immunoprecipitation (MeDIP) analysis was performed according to a previously study, with some minor changes [16]. Briefly, the genomic DNA was extracted from each muscle sample and then subjected to sonication to obtain fragments ranging in size from 200–800 bp. One microgram of the DNA fragments from each sample was denatured in a boiling water bath for ten minutes and then incubated with anti-5-methylcytosine (5 mC) antibodies (Abcam, Boston, MA, USA) on a rotator at 4 °C overnight. A portion of the DNA fragments (20 ng/μL) was preserved as the input. After immunoprecipitation, the DNA-antibodies complexes were captured with the protein A/G PLUS-agarose (Santa Cruz Biotechnology, Inc., Dallas, TX, USA) and digested with Proteinase K (Beyotime Biotechnology, Shanghai, China). The released methylated DNA fragments were then purified by phenol-chloroform extraction. The purified DNA and the input controls were subjected to real-time PCR to detect the methylation level of the target fragments. The information of the CpG islands in the promoter regions of different genes was predicted by using the CpG Finder and Urogene software (Beijing, China). The primers were designed with Primer Premier 5 software (Premier Biosoft, Palo Alto, CA, USA), and the sequences of the primers are listed in Table 2. The results were calculated as the degree of change relative to the input and are presented as the change degree relative to the mean of the control group.

### 2.4. Statistical Analysis

All statistical analyses were performed with SPSS 23.0 for Windows. All data are expressed as the mean ± standard error. One-way ANOVA followed by a post hoc test (LSD) were used to test the difference between the groups. The level of significance in all the analyses was set at *p* < 0.05.

## 3. Results

### 3.1. Betaine Alleviated LPS-Induced IL-6 Overexpression in Chicken Leg Muscle

The mRNA expression of the pro-inflammatory genes was detected in the leg muscle using real-time PCR. The mRNA expressions of IL-1β, IL-6, and tumor necrosis factor alpha (TNFα) were increased by 41-, 82-, and 13-fold 2 h after LPS treatment, respectively, compared with the control group (*p* < 0.01). In contrast, the chickens that drank water with betaine for two weeks showed a significantly alleviated IL-6 mRNA expression (*p* = 0.016) compared with the LPS-treated group. The expression level of IL-6 was decreased by about 40-fold. However, the mRNA expressions of IL-1β and TNFα were not affected by betaine consumption (Figure 1). 

### 3.2. Betaine Inhibited the LPS-Induced Down-Expression of TLR4 in Chicken Leg Muscle

As shown in Figure 2, the intraperitoneal injection of LPS decreased the mRNA expression of TLR4 (*p* < 0.05) and TLR2 (*p* < 0.01) but stimulated the mRNA expression of myeloid differentiation factor 88 (MyD88) (*p* < 0.01) in the broiler leg muscle. In contrast, the mRNA expression of TLR4 in the leg muscle of chickens that consumed betaine was considerably increased compared with that of the LPS group (*p* < 0.05), and it was comparable with that of the control group. However, betaine did not inhibit the effects of LPS on TLR2 and MyD88 mRNA expression.

### 3.3. Betaine Inhibited the LPS-Induced Overexpression of Tet2 in Chicken Leg Muscle

The effects of LPS on the mRNA expression of DNA methylation-associated enzymes were analyzed. As shown in Figure 3, LPS significantly decreased the mRNA expression of DNA methyltransferase 3a (DNMT3a) (*p* < 0.05) and *Tet1* (*p* < 0.05) and increased the expression of Tet2 (*p* < 0.01). In contrast, betaine inhibited LPS-induced Tet2 expression but did not inhibit the effect of LPS on the expression of DNMT3a and Tet1.

### 3.4. LPS Increased the Methylation Level in IL-6 Promoter

Two CpG islands were founded along 2000 bp in the 5′ flanking region of the IL-6 gene. The MeDIP-qPCR test showed that LPS significantly decreased the 5 mC level in the IL-6 promoter; however, betaine did not interfere with this effect (Figure 4). 

### 3.5. Betaine Blocks LPS-Induced Hyper-Methylation in the TLR4 Promoter

No CpG island was found along 2000 bp in the 5′ flanking region of the TLR4 gene. We analyzed the 5 mC level in the proximal (−44, −325) and distal end (−961, −1335) of this region. As shown in Figure 5, the LPS-stimulated chickens exhibited hypermethylation in the proximal end of the TLR4 promoter compared to the control group. In contrast, the 5 mC level in this region was significantly decreased in the Betaine+LPS group compared with the LPS group. 

## 4. Discussion

Skeletal muscle is the largest organ in the body and has also been identified as an endocrine organ. Cytokines and other peptides produced by skeletal muscle have been classified as “myokines”. IL-6 is the first identified and most studied myokine [26]. In skeletal muscle, IL-6 is lowly expressed under resting conditions, but it can be markedly induced during exercise [27]. In addition, skeletal muscle cells are capable of producing IL-6 in response to various stimuli such as LPS and inflammatory cytokines such as TNFα and IL-1β [28,29]. The present study found that the mRNA expressions of IL-6, IL-1β, and TNFα were remarkably increased 2 h after LPS stimulation, which is in line with the earlier study on mice that revealed that the peak expression of TNFα and IL-6 mRNA occurs 2 h after LPS intraperitoneal treatment. Considering the fold change of the expression of different cytokines, our results, together with those of the previous studies, showed that the mRNA expression of IL-6 is far more induced than that of TNFα and IL-1β [28,30], indicating that IL-6 may function as a main inflammatory factor during the acute stage of LPS-induced skeletal muscle inflammation.

It is known that stimuli such as LPS and a high fat diet engage TLR4/MyD88 signaling in skeletal muscle cells, activating the expression of IL-6 [28,31,32]. As a natural pattern recognition receptor of LPS, the mRNA expression of TLR4 was widely known to be increased after LPS treatment in various mammals and mammalian cells [33,34,35]. Different from these studies, here, we found that LPS decreased the mRNA expression of TLR4 and also TLR2. Similar with our results, Takahashi et al. found that TLR4 gene expression in the liver of male broilers intraperitoneally treated with LPS at a dose of 1.5 mg/kg BW was significantly lower than that of the control group during the observation period (2, 3, and 4 h) [36]. In addition, we also previously found that LPS intraperitoneal treatment can reduce TLR4 and TLR2 expression in broiler liver [37]. Another study on one-day-old broilers also showed that broilers stimulated intraperitoneally with LPS at a dose of 50 mg/kg BW exhibited significantly higher hepatic TLR4 at the early stage (6 h), which, however, remarkably decreased at the later stages (24 and 36 h) after the treatment [38]. These results indicated that chickens may show a different LPS responding pattern compared to mammals. LPS has been used to induce inflammation in chickens at variable doses ranging from as low as 0.1 μg/kg to as high as 50 mg/kg. A previous study demonstrated that chickens survived a mean dose of 517 mg/kg LPS [39]. On the contrary, mice that received a dose of about 4 mg/kg LPS showed a significant 50% death rate [40]. The different expression patterns of TLR4 after LPS stimulation may be attributed to different species combing with different LPS dosages. It is also predictable that the lower expression of TLR4 in chickens under LPS stimulation may contribute to the relatively higher resistance compared to the mammals.

Betaine’s anti-inflammation potential and its advantage of low pricing make it an appealing candidate for human nutrition and animal husbandry. Some studies have shown that betaine alleviates the inflammatory response induced by LPS through inhibiting the TLR4/MyD88/NF-κB signal transduction pathway [41]. Betaine can also inhibit high-mobility group box protein 1/TLR4 signal transduction, which may be one of the mechanisms of betaine in the treatment of liver injury in rats with non-alcoholic fatty liver disease [25]. In line with these previous studies, here, we found that betaine inhibited LPS-induced lower TLR4 expression and alleviated IL-6 expression. It is interesting that betaine only attenuated the mRNA level of IL-6, whilst having no effect on TNFα and IL-1β. In contrast, a previous study showed that, in vitro, betaine treating microglial cells decreased LPS-induced TNFα, IL-1β, and IL-6 levels in the media [41]. Considering that IL-6 was widely demonstrated to be the most highly induced pro-inflammatory cytokine in skeletal muscle, our result may be supportive of the hypothesis that betaine performs a cytokine-specific function facing inflammatory situations, though this is still not adequately proved. Baker et al. found that Resolvin E1 attenuated LPS-induced inflammation though specifically lowering the expression of IL-6 but not TNFα and IL-1β in C2C12 skeletal muscle myotubes [30], which supports the hypothesis that some ingredients can play an anti-inflammatory role by specifically targeting some of the pro-inflammatory cytokines.

The TLR4-triggered downstream signaling pathway is well characterized as controlling the expression of pro-inflammatory cytokines. Indeed, DNA methylation is highly imperative in regulating inflammatory genes. For example, epigenome-wide association studies linked DNA hypomethylation with increased inflammation [42]. A study carried out on intestinal epithelial cells found that cells with low TLR4 expression showed a significantly higher methylation level than cells with high TLR4 expression, which is associated with the different responsiveness to LPS stimulation [43]. In this study, we also found that LPS decreased the 5 mC level in the promoter region of IL-6 and increased the 5mC level in the promoter region of TLR4. As DNA methylation is widely known to be negatively related with the gene expression level, it is likely that DNA methylation is involved in the regulation of LPS-induced IL-6 and TLR4 expression.

In this study, the water supplementation of betaine did not alleviate LPS-induced IL-6 hypomethylation but inhibited LPS-induced TLR4 hypermethylation. Betaine consumption has been suggested to alleviate the LPS-induced general hypomethylation state of the body [44], whilst our results indicated that betaine consumption can also decrease the methylation level of some specific genes. Similar to our results, Hu et al. found that maternal betaine supplementation inhibited the mRNA expression of sterol regulatory element-binding protein 2 (SREBP2) by increasing the methylation level, but it contrarily increased the mRNA expression of cytochrome P450 family 7 subfamily A member 1 (CYP7a1) by decreasing the methylation level [8]. Studies carried out on mice also found that dietary betaine consumption can decrease the methylation of the peroxisome proliferator-activated receptor (PPAR) alpha promoter and then increase its expression [45], and betaine consumption through drinking water can decrease the microsomal triglyceride transfer protein (MTTP) methylation and increase its expression [21]. It is worth noting that DNA methylation machinery responds to methyl donors’ availability in a complex fashion; the methylation status of the promoters or regulatory regions of specific genes may respond differently from that of the global genomic DNA. 

The results of this study showed that the mRNA expression of Tet2 in leg muscle was significantly induced by LPS, and this effect was eliminated by betaine. Tet2 was found to play a crucial role in the initiation of inflammation through epigenetic regulation [46]. For example, a previous study showed that Tet2 can recruit to the promoter region of interferon regulatory factor 7(IRF7), maintaining the hypomethylation of IRF7 and then inducing its expression and the down-stream IFN expression in mouse plasmacytoid dendritic cells infected by herpes simplex virus [47]. Tet2 was also found to resolve inflammation by suppressing the expression of proinflammatory cytokines. Qin et al. found that Tet2 expression in mice and human macrophages were increased after LPS stimulation, and Tet2 was able to inhibit the expression of IL-1β, IL-6, and TNFα under LPS stimulation [48]. The results here found that the expression of Tet2 in skeletal muscle is soon increased after LPS treatment, which is coincident with the expression of proinflammatory cytokines. However, whether Tet2 functions as an initiator or a resolver of inflammation in chicken muscle induced by LPS was not determined. 

## 5. Conclusions

In summary, we found that LPS-induced skeletal muscle IL-6 excessive expression was alleviated by the consumption of betaine through drinking water. The mechanism may involve the epigenetic regulation of TLR4. Obviously, more in-depth investigations are needed to address whether other pro- or anti- inflammatory cytokines and signal transducers are involved in the betaine-mediated anti-inflammation process in chickens.

## Figures and Tables

**Figure 1 animals-12-01899-f001:**
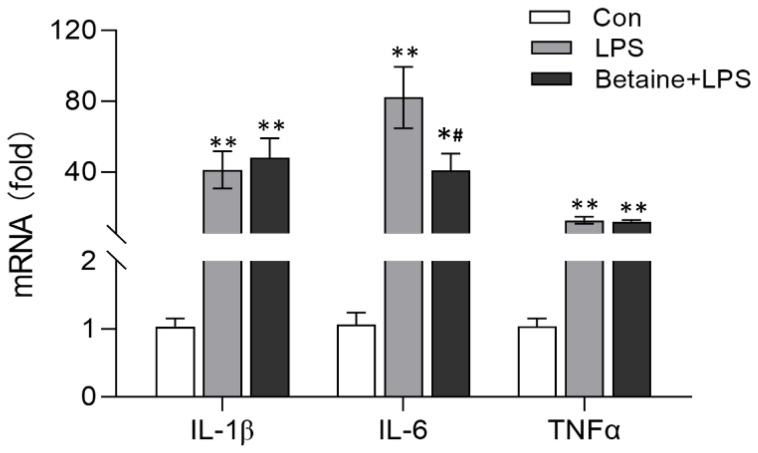
qRT-PCR quantification of the mRNA expressions of pro-inflammatory cytokines. Newly hatched broilers were provided with water supplemented with betaine at a dose of 1000 mg/L. At the age of two weeks, the broilers were intraperitoneally injected with LPS at a dose of 1 mg/kg BW. Two hours after the LPS treatment, leg muscle was collected, and the relative mRNA expressions of IL-1β, IL-6, and TNFα were determined using qRT-PCR. The treatment and method are the same for Figure 2 and Figure 3. Values are mean ± SEM, *n* = 6/group. Significance levels: * *p* < 0.05, ** *p* < 0.01 vs. control, ^#^ *p* < 0.05 vs. LPS.

**Figure 2 animals-12-01899-f002:**
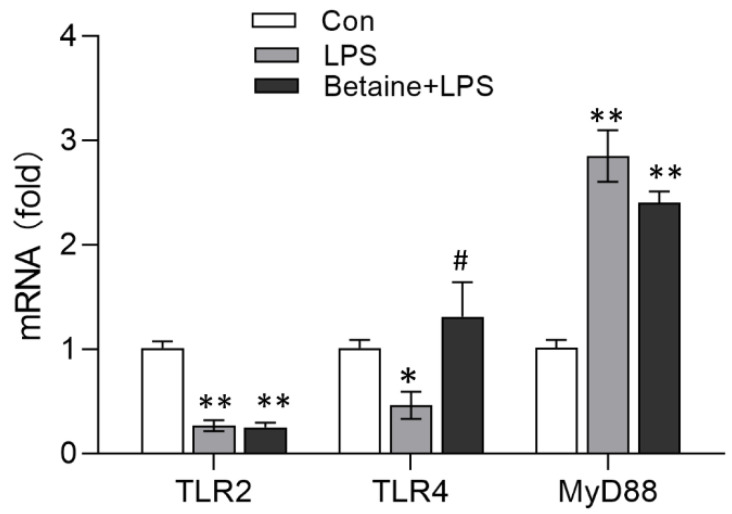
qRT-PCR quantification of the mRNA expression of TLR2, TLR4, and MyD88. The treatment and method were described in Figure 1 caption. Values are mean ± SEM, *n* = 6/group. Significance levels: * *p* < 0.05, ** *p* < 0.01 vs. control, ^#^ *p* < 0.05 vs. LPS.

**Figure 3 animals-12-01899-f003:**
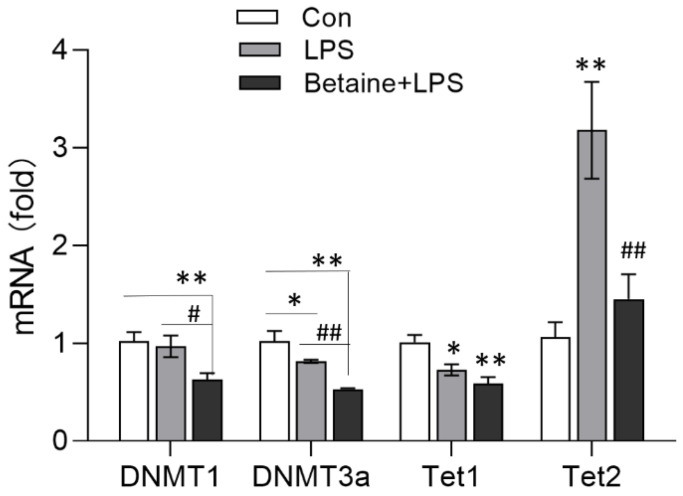
qRT-PCR quantification of the mRNA expression of DNA methylation-associated enzymes. The treatment and method were described in Figure 1 caption. Values are mean ± SEM, *n* = 6/group. Significance levels: * *p* < 0.05, ** *p* < 0.01 vs. control, ^#^ *p* < 0.05, ^##^ *p* < 0.01 vs. LPS.

**Figure 4 animals-12-01899-f004:**
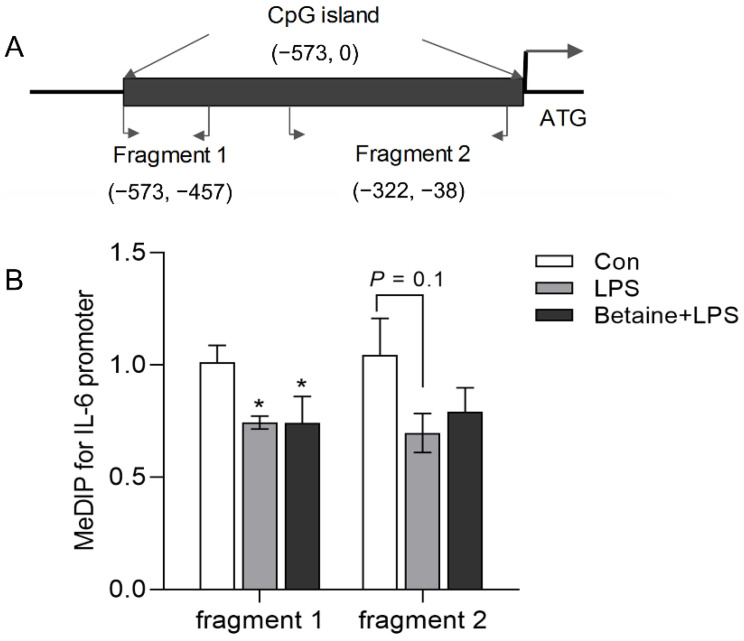
Schematic structure of the chicken Il-6 gene promoter and the methylation status of the CpG island. (**A**) The schematic structure of the chicken IL-6 promoter. █ shows the location of the CpG island. Fragment 1 and fragment 2 were regions detected by MeDIP-PCR. (**B**) The methylation levels of fragment 1 and fragment 2. Values are mean ± SEM, *n* = 4/group. Significance level: * *p* < 0.05 vs. control.

**Figure 5 animals-12-01899-f005:**
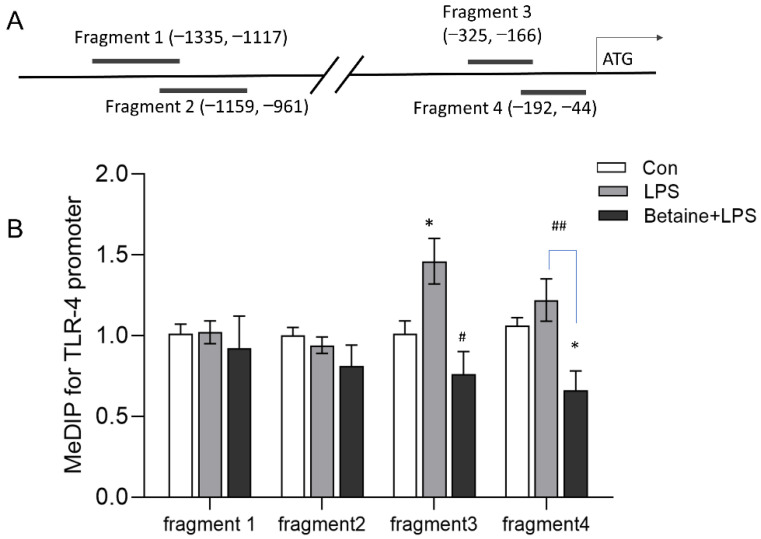
Schematic structure of the chicken TLR4 gene promoter and the methylation status. (**A**) The schematic structure of the chicken TLR4 promoter. Fragments 1-4 were regions detected by MeDIP-PCR. (**B**) The methylation levels of fragments 1–4. Values are mean ± SEM, *n* = 4/group. Significance levels: * *p* < 0.05 vs. control, ^#^ *p* < 0.05, ^##^ *p* < 0.01 vs. LPS.

**Table 1 animals-12-01899-t001:** Nucleotide sequences of the specific primers used in real-time PCR.

Target Gene	Sequence (F: Forward, R: Reverse)	GenBank Access
IL1-β	F: 5′-TTCCGCTACACCCGCTCACA-3′ R: 5′-TGCCGCTCATCACACACGAC-3′	NM_204524.2
IL-6	F: 5′-GAAATCCCTCCTCGCCAATCTG-3′ R: 5′-GCCCTCACGGTCTTCTCCATAAA-3′	NM_204628.2
TNFα	F: 5′-TCACCCCTACCCTGTCCCA-3′ R: 5′-AGCCAAGTCAACGCTCCTG-3′	NM_204267.2
TLR2	F: 5′-ATCCTGCTGGAGCCCATTCAGAG -3′ R: 5′-TTGCTCTTCATCAGGAGGCCACTC -3	NM_204278.1
TLR4	F: 5′-GTTTGACATTGCTCGGTCCT -3′ R: 5′-GCTGCCTCCAGAAGATATGC -3′	NM_001030693
MyD88	F: 5′-GTTTGATGCCTTCATCTGCTACT-3′ R: 5′-ATCCTCCGACACCTTCTTTCTAT-3′	NM_001030962.5
DNMT1	F: 5′-TTTTTTTACATAATCCTCCA -3′ R: 5′-AAAGTATCAATCCCCACTTG -3′	NM_206952.1
DNMT3a	F: 5′- ATCACCACTCGCTCCAACTC-3′ R: 5′-CCAAACACCCTCTCCATCTC-3′	NM_001024832.3
Tet1	F: 5′-AAAAGGAAGCGCTGTGAGAA-3′ F: 5′-CCACGCCAGTATGAGAATCA-3′	XM_015278732.1
Tet2	F: 5′-CGGTCCTAATGTGGCAGCTA-3′ F: 5′-TGCCTTCTTTCCCAGTGTAGA-3′	NM_001277794.1
β-actin	F: 5′-TGCGTGACATCAAGGAGAAG-3′ R: 5′-TGCCAGGGTACATTGTGGTA-3′	NM_205518

**Table 2 animals-12-01899-t002:** Primer sequences used in MeDIP-qPCR.

Target Gene	Sequence (F: Forward, R: Reverse, 5′—3′)	Product Size (bp)
IL-6fragment 1	F: GCGTGTGACGGCGTATAAC R: AAACCTCCTCGGGCTGGTG	116
IL-6fragment 2	F: GAGGCTGCCAGGCTCACCCCCC R: CCCTGAACGTGTATTTATCGAG	184
TLR4 fragment 1	F: GGTGTGTTTTCTGCTTGTGC R: GATGTTGGAGAGTTTGGGAG	218
TLR4 fragment 2	F: ACGCACTTTTTGTCTGCTGGC R: AGGAGATGGGCATGGGACTTC	188
TLR4 fragment 3	F: GTCTCCTCCAGAAACAATAGC R: CATCACATGAACACACACTCC	159
TLR4 fragment 4	F: GGTGCTGGAGTGTGTGTTC R: CGTGGTGTTGTATCGGTGT	144

## Data Availability

All relevant data are within the manuscript.

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
