# Peer review of "Betaine Alleviates LPS-Induced Chicken Skeletal Muscle Inflammation with the Epigenetic Modulation of the TLR4 Gene"

_animals, 2022, doi:10.3390/ani12151899_

Round 1

Reviewer 1 Report

The manuscript submitted by Guo et al evaluated the anti-inflammatory effects of betaine on LPS-induced muscle inflammation and the possible epigenetic mechanism. Authors found that betaine can alleviate LPS-induced muscle inflammation in chickens, and the regulation of aberrant DNA methylation might be a possible mechanism. I suggest the publication of this nice manuscript after minor revision.

1.      Please mention the full names of all abbreviations that mentioned for the first time

2.      Please provide the „Supplier, City, and Country“ for all materials and kits that used in this study

3.     Figures need to stand alone. this means that all necessary information to understand the Figures/tables have to be in each table, including the treatment description

4.      Line 94: Escherichia coli should be italic

5.      Please start the „ Introduction“ with a paragraph to describe the problem of inflammation if poultry and the need for Antiinflammatory. I suggest starting with the following paragraph „ Inflammation is the result of stress, regardless of the source or nature of the stress (biological, environmental, nutritional, physical, chemical, or psychological). Poultry production has experienced a major shift with the prohibition of the use of antimicrobials and antibiotics as growth promoters that may lead to an increase in the inflammatory level [https://doi.org/10.51585/gjvr.2021.3.0018].  Inflammation in return contributes to several pathologies such as skeletal muscle diseases which are characterized by the loss of skeletal muscle mass and reduction in poultry performance [Reference]

Author Response

Dear reviewer,

Thank you for your comments and valuable suggestions. Here are the details of all the changes we have made to the manuscript in response to the comments. All the changes are marked up using the “Track Changes” function

  1. Please mention the full names of all abbreviations that mentioned for the first time

Answer: All the abbreviations that were firstly mentioned in the manuscript have been rewritten in the form of full name.

  1. Please provide the “Supplier, City, and Country” for all materials and kits that used in this study

Answer: This information has been provided for all the materials and kits.

  1. Figures need to stand alone. this means that all necessary information to understand the Figures/tables have to be in each table, including the treatment description

Answer: We have added a brief description of the treatment and method to each of the figure. Considering that the treatments and methods used in Figure 1-3 are same, we just added this information in figure 1, and used “Treatment and method are same for figure 2 and 3” to guide the readers.

  1. Line 94: Escherichia coli should be italic

Answer: Thank you for reminding us this. We have revised it.

  1. Please start the “Introduction” with a paragraph to describe the problem of inflammation if poultry and the need for anti-inflammatory. I suggest starting with the following paragraph “Inflammation is the result of stress, regardless of the source or nature of the stress (biological, environmental, nutritional, physical, chemical, or psychological). Poultry production has experienced a major shift with the prohibition of the use of antimicrobials and antibiotics as growth promoters that may lead to an increase in the inflammatory level [https://doi.org/10.51585/gjvr.2021.3.0018]. Inflammation in return contributes to several pathologies such as skeletal muscle diseases which are characterized by the loss of skeletal muscle mass and reduction in poultry performance [Reference]

Answer: Thank you so much for your suggestion. We have restarted the first paragraph of the Introduction with the sentences you provided above. We did not change a single word as what you offered is so perfect!

Reviewer 2 Report

They have provided multiple P-values in the paper.  However, most of them are represented as inequalities, like "P<0.01". When multiple P-values are reported in the literature, these must be corrected with considering multiple comparison correction. The authors must provided actual P-values and evaluate them with considering multiple comparison corrections. Before performing these evaluations, I cannot judge the contents, since all of P-values might not be significant after the correction.   

Author Response

Dear reviewer,

Thank you for your comments and valuable suggestions. We gave the response to the comment below.

 There were three groups in this experiment. We used the One-way ANOVA followed by a post-hoc test (LSD) method to detect differences between groups. LSD (least significant difference) is one of commonly used multiple comparison test. We have mentioned this method in the manuscript, please see line 143-146. This statistical analysis method and the expression of P value as P<0.01/0.05 have been widely used, and many studies published in Animals also use this method.

Reviewer 3 Report

The manuscript is well written and well presented. Suitable for publication in current form. 

Author Response

Thank you for your support!

Reviewer 4 Report

Dear Authors

The article contains interesting information and novelty elements and is in line with the current research trend in this field. The manuscript was prepared according to the guidelines. However, in my opinion, it needs to be revised before possible acceptance. I present my remarks and comments in the attachment.

Author Response

Dear reviewer,

Thank you for your comments and valuable suggestions. We responsed to all the comments in the attachment. And the details of all the changes we have made to the manuscript in response to the comments. All the changes are marked up using the “Track Changes” function.
